# AAA+ ATPases in Protein Degradation: Structures, Functions and Mechanisms

**DOI:** 10.3390/biom10040629

**Published:** 2020-04-18

**Authors:** Shuwen Zhang, Youdong Mao

**Affiliations:** 1Center for Quantitative Biology, School of Physics, Peking University, Beijing 100871, China; swzhang@pku.edu.cn; 2Dana-Farber Cancer Institute, Harvard Medical School, Boston, MA 02138, USA

**Keywords:** AAA+ ATPase, ATP-dependent proteolysis, substrate translocation, 26S proteasome, Cdc48/p97, mitochondrial protease

## Abstract

Adenosine triphosphatases (ATPases) associated with a variety of cellular activities (AAA+), the hexameric ring-shaped motor complexes located in all ATP-driven proteolytic machines, are involved in many cellular processes. Powered by cycles of ATP binding and hydrolysis, conformational changes in AAA+ ATPases can generate mechanical work that unfolds a substrate protein inside the central axial channel of ATPase ring for degradation. Three-dimensional visualizations of several AAA+ ATPase complexes in the act of substrate processing for protein degradation have been resolved at the atomic level thanks to recent technical advances in cryogenic electron microscopy (cryo-EM). Here, we summarize the resulting advances in structural and biochemical studies of AAA+ proteases in the process of proteolysis reactions, with an emphasis on cryo-EM structural analyses of the 26S proteasome, Cdc48/p97 and FtsH-like mitochondrial proteases. These studies reveal three highly conserved patterns in the structure–function relationship of AAA+ ATPase hexamers that were observed in the human 26S proteasome, thus suggesting common dynamic models of mechanochemical coupling during force generation and substrate translocation.

## 1. Introduction

Protein degradation plays a fundamental role in the maintenance of cellular homeostasis and the regulation of nearly all major cellular processes, such as cell cycle regulation, gene expression, signal transduction, immune response, apoptosis and carcinogenesis [1]. Proteolysis affects not only misfolded or otherwise damaged proteins, but also regulatory proteins to maintain the function of cellular integrity, thereby preventing human diseases such as cancer and neurodegenerative diseases [2,3,4,5]. Many drugs are developed for the treatment of these diseases by targeting key proteases or regulators in corresponding proteolytic pathways. For example, the dipeptide proteasome inhibitor Bortezomib was approved by the US Food and Drug Administration (FDA) in 2003 for treating multiple myeloma [6,7]. Understanding the structures and functions of these key protease complexes and their implications in pathological conditions is therefore instrumental for therapeutic development.

Proteins that are earmarked for degradation are usually well-folded and are therefore tagged via ubiquitylation to become distinguishable from normal cellular constituents. The globular domains of these proteins have to be unfolded and delivered to the proteolytically active sites before they can be broken down into short polypeptides. This sophisticated task is thought to be carried out by protease complexes containing ring-like structures assembled from adenosine triphosphatase (ATPase) of the ATPases associated with a variety of cellular activities (AAA+) superfamily that is essential for most proteolytic activities [8,9]. Thus, proteolytic subunits or domains must assemble with AAA+ ATPases into an ATP-fueled protease complex machinery that couples ATP hydrolysis with consecutive proteolysis reactions [10,11,12]. The proteolytic subunits typically assemble into a cylinder-shaped chamber, such as the proteasome core particle (CP) [10,11,13,14], HslV [15] and ClpP [16]. The AAA+ ATPase subunits function as substrate-remodeling engines or motors in an ATP-dependent manner, and assemble into a hexameric ring with a central axial pore that guards the entry port of the proteolytic chamber [11]. In the 26S proteasome, a heterohexameric AAA–ATPase ring associates with the lid subcomplex to form a proteasomal regulatory particle (RP) [10,11]. Powered by the AAA+ ATPase engines, these protease machines can recruit and unfold substrates carrying specific degradation signals and translocate them into the proteolytic chamber for breakdown.

On the basis of sequence and structural comparison, the AAA+ superfamily members have been categorized into a number of AAA+ “clades”, where the clade is defined as a branch evolved from a common ancestor and consists of several distinguishing protein families [17]. They are characterized by different insertions of special sequences or secondary structural elements at specific regions within core AAA+ domains. Common structural properties and evolutionary classification of these clades have been reviewed in detail elsewhere [18,19,20,21]. In the case of protease complexes, AAA+ ATPases can be divided into two clades. One is called the classic AAA clade, including the proteasomal family, FtsH family, Cdc48 family, and ClpA/B/C-Domain 1 (D1) family. The other clade is called HCLR, which features a special insertion sequence called pre-sensor 1 residing in at least one AAA+ module, including the HslU/ClpX family, ClpA/B/C-Domain 2 (D2) family, and Lon family. Besides features specific to the clades, these subfamilies also exhibit different domain architectures [22] (Figure 1a). For example, the protease domain and AAA+ module that consists of a large and small AAA subdomain, are expressed in the same protein subunit for FtsH and Lon, whereas these two kinds of modules reside in distinct subunits in other protease complexes. Cdc48, HslU and ClpA/B/C even contain two AAA+ modules per subunit. In recent years, cryogenic electron microscopy (cryo-EM) studies in combination with biochemical experiments have elucidated the structures and mechanisms of several protease complexes in the act of unfolding or degrading a substrate protein. In this review, we will focus on these recent results and summarize the structure, function and working mechanisms of protease AAA+ ATPases.

## 2. AAA+ ATPases in Protein-Degradation Machinery

### 2.1. AAA+ ATPases in the 26S Proteasome

The ubiquitin–proteasome system (UPS) and autophagy–lysosomal pathway provide the majority of the intracellular protein degradation activities in eukaryotic cells [1,3,23]. In the UPS, substrates modified by polyubiquitin chains are selectively targeted and destructed by the 26S proteasome, a 2.5-megadalton proteolytic molecular machine equipped with AAA ATPases. As the endpoint of the UPS, the 26S proteasome is the most sophisticated protease complex known, ubiquitously found in all eukaryotes [24,25]. There have been some excellent reviews covering proteasome structure and function [23,26,27,28], assembly [29,30], ubiquitin recognition [31,32,33], and proteasomal deubiquitinating enzymes [34].

The 26S proteasome holoenzyme assembles from one CP and two RPs capping both sides of the CP cylinder (Figure 1b and Figure 2a). Proteasomal CP, also known as the 20S proteasome, is composed of distinct α-type and β-type subunits that are stacked into a barrel-like α_7_β_7_β_7_α_7_ assembly. RP, also known as 19S or the PA700 complex, is the most commonly found proteasome activator whose assembly and function is dependent of ATP. Other ATP-independent activators, such as 11S (PA28) and Blm10 (PA200), can also be associated with the CP to activate the proteasome holoenzyme [35].

The RP can be structurally divided into a lid subcomplex, which comprises nine regulatory particle non-ATPase (RPN) subunits (RPN3, RPN5, RPN6, RPN7, RPN8, RPN9, RPN11, RPN12, and RPN14/DSS1/Sem1; Table 1), and a base subcomplex, which consists of RPN1, RPN2, RPN13 and six paralogous, distinct regulatory particle ATPase (RPT) subunits (RPT1–RPT6) from the classic AAA family. Another RP subunit, RPN10, interacts with both the base and lid and was previously considered part of the base. Each RPT subunit consists of a N-terminal helical domain, which dimerizes into a coiled coil (CC) between adjacent subunits (RPT1/RPT2, RPT6/RPT3, and RPT4/RPT5), an oligonucleotide- and oligosaccharide-binding (OB) domain, and a C-terminal AAA domain, which comprises a large and small AAA subdomain (Figure 2d). Like other AAA+ ATPases, the AAA domain of RPT contains highly conserved motifs, including Walker A, Walker B, sensor 1, arginine finger (R-finger), sensor 2, and pore-1/2 loops [17,18,19,20,21,22]. The pore loops interact with substrates directly and form a central translocation channel in a right-handed spiral staircase arrangement. The nucleotide-binding pocket of each RPT subunit is surrounded by Walker A, Walker B, sensor 1, and sensor 2 motifs from one RPT subunit, working in “cis”, and two R-finger motifs from the large AAA subdomain of the clockwise neighboring RPT subunit (viewed from the side of the OB ring, Figure 2e, left), functioning in “trans” and allowing for allosteric communication between adjacent subunits [36]. With these structural motifs, ATP binding and hydrolysis drive conformational changes of the ATPases, and the chemical energy of ATP hydrolysis is converted into the mechanical work of substrate translocation through the axial channel. Mutagenesis experiments have found that each of the six ATPases exhibited functional asymmetry in substrate degradation although they all share those conserved motifs [37,38].

Recognition of a ubiquitylated substrate, the first step of substrate processing, is mediated principally by the ubiquitin receptors, such as RPN1 [39], RPN10 [40,41], and RPN13 [42,43]. After substrate recruitment, the flexible initiation region of the substrate is then captured by the pore loops of the RPT subunits [11,32,44,45]. To allow subsequent degradation, conjugated ubiquitin chains are removed by either the intrinsic deubiquitinase (DUB) subunit RPN11 [11,46,47,48] or the auxiliary DUBs like USP14/Upb6 [49] and UCH37/UCHL5 [50]. The globular domains of a substrate are then mechanically unfolded and translocated through the narrow axial channel of the heterohexameric RPT ring. The central entry port of the CP proteolytic chamber (also known as the CP gate) remains closed before activation by the RP [51,52,53]. Opening of the CP gate is triggered by docking C-terminal tails of the RPT subunits into the inter-subunit surface pockets of the α-ring (also known as α-pockets), which are formed between neighboring α-subunits [54,55]. Structural studies on the human 26S proteasome have established that insertion of the C-termini of RPT3 and RPT5 into α-pockets, which contain conserved hydrophobic-Tyr-X (HbYX) motifs, ensures RP’s association with CP but is not sufficient for CP gate opening [51,52,53]. For the yeast 26S proteasome, the insertion of the Rpt2 C-terminus also seems to contribute to the complex assembly in addition to those of Rpt3 and Rpt5 [56,57,58]. In both the human and the yeast proteasome, CP gate opening was observed when the C-termini of all RPT subunits except RPT4 are engaged with the α-pockets [11,51,56,58,59].

### 2.2. AAA+ ATPases in Cdc48/p97

Cdc48 in yeast, and its ortholog p97 or valosin-containing protein (VCP) in higher eukaryotes, remodel ubiquitinated substrates for ubiquitin-dependent degradation [60,61,62,63], and play an important role in the UPS pathway [64], especially in endoplasmic reticulum (ER)-associated protein degradation (ERAD) [65,66,67,68] and outer mitochondrial membrane associated degradation (OMMAD) [69,70,71]. It has been shown that archaeal Cdc48 can artificially assemble with the 20S proteasome through in vitro crosslinking [72]. Since the 26S proteasome requires an unstructured polypeptide segment in its substrate to initiate processing [44,73,74], Cdc48 can act upstream of the proteasome when the substrate is well-folded, without a flexible initiation region, or located in membranes [75,76]. Cdc48 can partially or completely unfold the substrate, and transfer it to the 26S proteasome for degradation with the assistance of the shuttling factors Rad23 or Dsk2 [77,78,79], or directly into the 20S complex [80,81]. Furthermore, Cdc48 is also involved in other cellular processes, including autophagy [82,83], ribosomal quality control [84,85], extraction of chromatin-bound proteins [86,87,88], membrane fusion and vesicular trafficking [89,90]. It is not only associated with several diseases [91,92,93], but has also been identified as a promising anti-cancer drug target due to its general role in protein quality control, homeostasis and cell viability [94,95,96,97].

Similar to the proteasomal RP, AAA+ ATPases in the Cdc48 complex can unfold and translocate a substrate through its central pore. In addition to an N-terminal (N) domain and a flexible C-terminal tail, a Cdc48 monomer encompasses two tandem ATPase domains (D1 and D2), each forming a ring-like homohexamer in the complex (Figure 1a,c). Both D1 and D2 are homologous to the single AAA domain of proteasome-activating nucleotidase (PAN) and proteasomal RPT subunits, hosting a nucleotide-binding pocket and pore-1/2 loops that can interact with substrates [17,18,19,20,21,22,98]. Thus, one Cdc48 complex possesses twelve ATP-binding sites in total. The N domain plays an important role in binding cofactors like heterodimer Ufd1/Npl4, and its position is not fixed with respect to the double ring, depending on the nucleotide-binding states of the D1 domain. Upon ATP binding to the D1 ATPases, N domains are displaced from a “downward conformation” coplanar with the D1 ring to an “upward conformation” above the D1 plane, regulating the cofactor binding via this conformational change [99,100,101,102,103]. To engage a substrate, Cdc48 often needs the help of various cofactors, which usually bind to the N domains or the extreme C-termini of Cdc48 for substrate recognition, ubiquitin chain modification, and fine-tuning of substrate processing [104,105]. These cofactors supply Cdc48 with the substrate specificity and pathway selectivity.

### 2.3. AAA+ ATPases in FtsH-Like Mitochondrial Proteases

Separated by two phospholipid bilayers, most of the mitochondrial proteins (as well as chloroplastic proteins [106,107]) cannot be accessed by the cytosolic UPS. Instead, they rely on independent proteolytic pathways within the mitochondria that are distinct from their counterpart in the cytosol [108,109]. The mitochondrial proteolysis system is crucial for a number of cellular processes that are essential for maintaining mitochondrial functions and homeostasis, such as apoptosis, mitochondrial biogenesis, and stress responses [110]. Similar to the 26S proteasome, numerous mitochondrial proteases comprise ring-like hexamers of AAA+ ATPases, which can be genetically traced to ancestral bacterial enzymes and are categorized into three highly conserved protease families: the Lon proteases, the Clp proteases, and FtsH-like AAA proteases [111]. The former two families are localized in the mitochondrial matrix space, whereas the FtsH-like protease family are uniquely membrane-anchored metalloproteases, embedded within the inner membrane (IM) of mitochondria, with their catalytic sites exposed either to the matrix or the intermembrane space (IMS), referred to as m- or i-AAA proteases, respectively. The structural and molecular basis for the functional specialization of FtsH-like proteases remains poorly understood [108,109,112]. Independently of the different intracellular localization, the FtsH-like proteases share a common structural topology that comprises an N-terminal domain, an AAA+ ATPase domain, and a zinc–metalloprotease domain and a transmembrane domain (Figure 1a,d). Both AAA+ and protease domains arrange as a hexameric ring and vertically stack together around a central pore. In eukaryotes, the i-AAA protease is composed of six identical subunits (YME1L in mammals, or Yme1 in yeast). By contrast, the m-AAA protease is assembled by distinct subunit compositions with multiple isoforms. In mammals, it can either form a homohexamer of AFG3L2 subunits or a heterohexamer of alternating AFG3L2 and paraplegin (SPG7) subunits [113,114], whereas in yeast it is an obligate heterohexamer of alternating Yta10 and Yta12 subunits [115].

## 3. Principal Working Mechanisms of AAA+ ATPases

### 3.1. Conformational Changes of AAA ATPases in the 26S Proteasome

The structures of the 20S proteasome (CP) in archaea and yeast were determined by X-ray crystallography more than two decades ago [13,14]. In contrast to the high stability of the CP, the RP, and particularly the AAA–ATPases module, exists as a highly dynamic component, sampling an extensive conformational landscape both in its free form and in the context of the 26S holoenzyme [11,51,59,116,117,118]. Recent studies of single-particle cryo-EM and cryo-electron tomography (cryo-ET) suggested that in vivo the 26S proteasome holoenzymes spontaneously sample distinct alternative conformational states, and mostly stay in a basal resting state in the presence of ATP and the absence of a substrate [51,117,119,120]. Several studies found that the conformational distributions of the 26S proteasome can be modified by using hydrolysis-inactivated Walker B mutations [58], deactivating certain subunits with mutations or inhibitors [120,121], or by replacing ATP with the slowly hydrolyzed ATPγS or nonhydrolyzable ATP analogs [56,57,59,117,122]. At least six distinct conformations of both human and yeast proteasomes without any substrate bound have been observed [26]. Since the proteolysis process was absent, the observed conformational changes mainly reflect an idle ATPase motor with no external work output.

Recent cryo-EM studies offer the first high-resolution views of dynamic substrate interactions with the 26S proteasome and insights into the inner workings of this macromolecular machine [11,118]. Unlike several previous studies that completely replaced ATP with ATPγS or nucleotide analogs [56,58,59,122,123], Dong et al. first primed the substrate-engaged proteasome with ATP, then diluted ATP with ATPγS after the initial phase of substrate engagement with the human 26S proteasome in a time-dependent manner to decelerate the hydrolysis activity of AAA+ ATPases, which is expected to maximize the conformational diversity and heterogeneity of the 26S proteasome being captured [11]. To compensate for the complexity of cryo-EM analysis conferred by the extreme conformational heterogeneity, the researchers collected an unusually large cryo-EM dataset, extensively using the latest machine-learning tools in data clustering [124,125,126,127] and eventually obtained seven conformational states of the substrate-bound human 26S proteasome at 2.8–3.6 Å resolution [11]. By contrast, de la Peña et al. inactivated the yeast DUB Rpn11 with the inhibitor ortho-phenanthroline and reported four conformations of substrate-bound yeast 26S proteasome at 4.2–4.7 Å resolution [118]. Although the inhibition of Rpn11 may have created structural features that are either potentially off-pathway or physiologically less relevant, the AAA–ATPase ring structures in the major conformations of the substrate-engaged yeast 26S proteasome showed highly comparable features, within the limit of their resolutions, to those in states E_D1_ and E_D2_ of the human counterparts, indicating a high degree of conservation of the underlying structural mechanisms for substrate processing by the proteasome across all eukaryotic kingdoms [11,118]. Together, these structural snapshots depict a spatiotemporal continuum of polyubiquitylated substrate degradation dynamics, shedding light on the complete cycle of substrate processing by the 26S proteasome, from initial ubiquitin recognition [11], deubiquitylation [11], and translocation initiation [11], to processive substrate degradation [11,118].

The conformational changes of RPT subunits in the proteasome appear to be strongly coupled with all major steps of substrate processing, and play an important role in dynamic regulation of the proteasome function [11]. The overall structural relationships between RP and CP and between the lid and base seem to be highly consistent among the 26S proteasome conformations with or without substrates. One of the key conformational states, termed “state E_B_” representing the human proteasome at the deubiquitylation step, appears to be missing in all previous studies except for one [11]. The structure of the 26S proteasome in state E_B_ reveals an unexpected quaternary subcomplex involving RPN11, RPN8 and RPT5 (Figure 2b). Around the scissile isopeptide bond between the RPN11-bound ubiquitin and the substrate lysine, a ternary interface is formed between RPN11, RPN8 and the N-loop of RPT5 which emanates from the top of its OB domain to efficiently carry out the deubiquitylation step (Figure 2b) [11]. Within the axial channel of the RPT ring, the substrate is in contact with the tyrosine or phenylalanine residues of pore-1 loops, where the aromatic side chains intercalate with the zigzagging mainchain of the substrate polypeptide through hydrophobic interactions. Furthermore, the N-terminal CC domains of RPT subunits, in contact with the lid subcomplex, allosterically regulate ATPase activity in a long-range fashion and contribute to the conformational switch of the holoenzyme [11,128]. The RP undergoes dramatic rotation (30°–40°) and translation relative to the CP during the transition of the CP gate from its closed to its open state (Figure 3a). The observation of state E_B_ provides information about critical intermediates missed in all other studies [51,52,53,59,118], as it allows the observation of the gradual activation of the CP by the stepwise insertion of RPT C-tails into the α-pockets (Figure 3b) [11]. During the process of the CP gate opening, the relative position of the catalytic site of DUB RPN11, the translocation channel of the RPT ring, and the gate of CP are all gradually aligned coaxially [11,118,121,122], so that the substrate polypeptide can be threaded progressively into the proteolytic CP chamber.

### 3.2. How Does a Substrate Interact with the ATPase Hexamer?

In each RPT subunit, there are two key structural motifs predominantly coupling the ATP hydrolytic cycle with substrate processing [17,18,19,20,21,22]. One is the pore loop, including pore-1 and pore-2 loops, both heading towards the inner channel [37,38,129,130] (Figure 2c). Substrate-contacting pore-1 loops of six RPT subunits are almost evenly distributed along the unfolded substrate polypeptide like a spiral staircase, with two adjacent pore-1 loops spanning two amino acid residues in the substrate, presumably corresponding to a one-step translocation driven by hydrolysis of a single ATP molecule [11]. This “two-residue spacing” of substrate-contacting pore-1 loops appears to be a key structural feature that is highly conserved among many AAA+ ATPases, including the 26S proteasome [11,118], Cdc48/p97 [76,131], FtsH-like AAA proteases [132,133], and Hsp104 disaggregase [134]. This suggests a conserved mechanism underlying the force generation by nonspecific, intercalated stacking interactions between the pore-1 loop’s aromatic residues and the substrate sidechains. The pore-2 loops form a similar but shorter staircase underneath the pore-1 loops, supporting the opposite side of the substrate through charged acidic residues [11,51,59,118]. Gripped by pore loops, the substrate may be propelled towards the proteolytic chamber by the ATP-fueled rigid-body movements of the RPT subunits.

### 3.3. How Are Substrate Interactions Coupled with ATP Hydrolysis?

A key structural motif in the AAA domain is the nucleotide-binding pocket near a hinge-like short loop, connecting the large and small AAA subdomains (Figure 2d). An ATP molecule within this pocket contacts both AAA subdomains, thus determining the hinge configuration and locking the AAA domain into a single rigid body. When ATP is hydrolyzed and γ-phosphate and ADP are subsequently released, the corresponding ATPase subunit undergoes an outward flip of 30°–40°, resulting in the disengagement of the ATPase from the substrate (Figure 2e). Therefore, the chemical states of nucleotides in the pocket determine the conformational state of the AAA domain. In the cryo-EM maps of the human 26S proteasome, three states of the nucleotide-binding pocket were observed: ATP-bound, ADP-bound, and an apo-like state in which only a very weak or partial density is present inside its nucleotide-binding pocket. Systematic structural alignment of ATPases among different states showed that ATP binding or ADP release leads to a generic hinge-like rotation of 15°–25° between its small and large AAA subdomains [11]. By contrast, the release of γ-phosphate after ATP hydrolysis appears to be insufficient to immediately trigger an intrinsic motion in the AAA domain, but instead converts the chemical energy harvested from ATP hydrolysis into an intra-domain potential energy stored in the AAA domain. Under this circumstance, the whole AAA domain may still rotate as a rigid body upon inter-subunit interactions. The subsequent release of ADP liberates the stored potential energy and converts it into kinetic energy of hinge-like rotation between the large and small AAA subdomains, which disengages the corresponding ATPase subunit from the substrate, and spread the kinetic energy out to drive the rigid-body rotation of four or five substrate-bound ATPase subunits of the holoenzyme that propels the substrate forward [11,118]. These structural findings are more compatible with a sequential ATP hydrolysis model rather than a random one in the ATPase hexameric ring (Figure 2e, 4), which is in line with biochemical studies of the proteasome and the bacterial Clp protease [135,136,137]. One of the notable features of this model is that not all ATPase subunits make contact with the substrate simultaneously; instead, at least one subunit is disengaged from the substrate upon ADP release (Figure 3c). This appears to be a common feature observed in most substrate-bound ATPase hexamer structures [131,132,133,134,138,139,140,141,142].

### 3.4. How Is the Cycling of ATP Hydrolysis Coordinated for Functional Regulation?

The ATPase ring in the 26S proteasome processes more inter-subcomplex interactions than most of other AAA+ ATPase complexes. The ATPase ring forms many inter-subunit interfaces with the lid subcomplex and the RPN subunits in the base, and it forms a multivalent, highly dynamic interface with the α-ring in the CP, mostly via the C-terminal tails of RPT subunits. The structural complexity is presumably evolved to accommodate the functional complexity of the proteasome in ubiquitin recognition, deubiquitylation and substrate unfolding. Unexpectedly, three distinct modes of coordinated ATP hydrolysis in the proteasomal ATPase ring have been discovered to regulate the key functional steps of the proteasome [11] (Figure 4). The ability to function in multiple modes via the same AAA–ATPase hexamer suggests the existence of multiple pathways of conformational changes induced by coordinated ATP hydrolysis and inter-subcomplex interactions (lid-base and RP–CP interactions) [11,51,59].

*Mode 1*: this features coordinated ATP hydrolysis in a pair of oppositely positioned ATPases and was observed in states E_A1_, E_A2_ and E_B_ of the human 26S proteasome, corresponding to the intermediate steps of initial ubiquitin recognition and deubiquitylation [48,143]. Before the proteasome in state E_B_ gets ready to remove the ubiquitin chain from the substrate with the DUB RPN11, the ADP bound to RPT6 in state E_A_ is released. Meanwhile, the ATP in both RPT2 and its opposite subunit RPT4 are hydrolyzed. These events drive the outward rotation and partial refolding of RPT6 and an iris-like movement in the AAA ring that opens its axial channel for initial substrate insertion into the axial channel. This mode of coordinated ATP hydrolysis was also observed in the crystal structure of the hexameric ClpX protease [144], which drives rather different conformational changes in the ATPase ring compared to those in the 26S proteasome, likely because of the lack of a substrate.

*Mode 2*: this features coordinated ATP hydrolysis in at least two adjacent ATPases and was observed in states E_C1_ and E_C2_ of the human 26S proteasome, corresponding to the intermediate steps of CP gating and initiation of substrate translocation. Another key structural feature accompanying this mode is the simultaneous disengagement of at least two adjacent ATPases from the substrate. After deubiquitylation, the proteasome transforms from state E_B_ to E_C1_ to initiate substrate translocation and prepare for the allosteric regulation of the CP gate opening. This conformational switch is achieved when ATP molecules in two adjacent subunits, RPT1 and RPT5, are hydrolyzed, with RPT1 and its clockwise-neighboring subunit, RPT2, disengaged from the substrate. As RPT6 binds ATP again and returns to the top of the substrate-bound pore loop staircase, the substrate is translocated forward by a distance of two residues, termed “one step”. However, during the following E_C2_-to-E_D1_ transition, both the substrate-disengaged RPT1 and RPT2 need to bind ATP and return to the top of the substrate-bound pore-loop staircase, while RPT5 is about to release its ADP, which together drives a two-step forward translocation of the substrate.

*Mode 3*: this features coordinated ATP hydrolysis in only one ATPase at a time and was observed in states E_D1_ and E_D2_ of the human 26S proteasome corresponding to the intermediate steps of processive substrate unfolding and translocation. When a pore-1 loop reaches the CP-proximal position, the bottom of the staircase, this RPT subunit is always ADP-bound. Then the ADP molecule is released from the binding pocket, and the subsequent hinge-like rotation between the small and large AAA subdomains disengages the pore loop from the substrate and flips this RPT subunit outwards, away from the ATPase ring. At the same time, the ADP-bound anticlockwise-adjacent RPT subunit is pushed to the bottom of the substrate-bound pore-loop staircase; and its clockwise neighboring RPT subunit, which was in an apo-like state and detached earlier, now acquires a new ATP and reengages with the substrate at the top of the staircase through a hinge-like rotation in a “hand-over-hand” fashion. Contemporaneously with these concerted motions, the other three subunits that are engaged with the substrate are mostly ATP-bound and rotate downwards as a rigid body driven by the conformational changes in the RPT subunits undergoing nucleotide exchange. Although only one step of the substrate translocation has been observed in *Mode 3* [11,118], from the study on the proteasome-activating nucleotidase (PAN) proteasome, an archaea homolog of the 26S proteasome, five distinct conformations in the PAN ATPase ring have been detected in the absence of a substrate. In each conformation, only one ATPase was disengaged from the rest of the ATPase ring, a key feature consistent with *Mode 3* [145]. However, due to limited resolutions (~4.9 Å) and lack of structural features labeling the time sequence in the cryo-EM reconstructions, it is yet to be confirmed whether ATP hydrolysis only occurs in one ATPase at a time, followed by *Mode-3* hydrolysis in the adjacent subunit occurring repetitively around the ring [145]. Such a coordinated sequential hydrolysis model suggests a unidirectional propagation of conformational changes in the ATPase ring.

### 3.5. Is There Real Evidence for a Sequential Model of Coordinated ATP Hydrolysis?

Because the majority of substrate-bound AAA+ ATPase structures were solved in only one conformation at high resolution as per their biochemical condition, the sequential “hand-over-hand” model of coordinated ATP hydrolysis around the ATPase ring is largely speculative and hypothetical [131,132,133,138,146]. In few studies where coexisting conformations were obtained, there were no intrinsic features that revealed the time sequence of the events along the pathway of chemical reactions [118,145,147]. One exception exists in the high-resolution cryo-EM reconstructions of the substrate-bound human 26S proteasome that contain inherent features of ubiquitin densities verifying the time sequence of the corresponding states of chemical reactions [11]. They are also the only available set of atomic structures showing a complete cycle of ATP hydrolysis regulating all six ATPases around the heterohexameric ring. Notably, states E_A1_ and E_A2_ show ubiquitin densities near RPN10 and RPN11 and no substrate density inside the AAA ring, whereas state E_B_ shows density features of both RPN11-bound ubiquitin and AAA-bound substrate with a visible isopeptide bond between the ubiquitin and substrate. In contrast, state E_C1_ shows densities of both RPN11-bound ubiquitin and AAA-bound substrate, with the isopeptide bond being completely absent in density, unambiguously verifying that this state is chemically post-deubiquitylation following state E_B_. Interestingly, state E_C2_ shows virtually identical ATPase conformation but lacks the RPN11-bound ubiquitin and the nucleotide density in RPT1, thus verifying that it represents the state immediately after state E_C1_. Both states E_D1_ and E_D2_ show no RPN11-bound ubiquitin density but exhibit an open CP gate. Along with other detailed dynamic features, such as gradual opening of the CP gate and conformational changes of the RP, these structures suggest a spatiotemporal continuum providing us with the only direct evidence for sequential ATP hydrolysis in a counterclockwise direction around the proteasomal ATPase ring [11]. One should note, however, that the observed complete cycle of ATP hydrolysis navigating the ATPase ring reflects a mixture of *Modes 1*, *2* and *3* [11]. This leaves the possibility open that a sequential *Mode 3* hydrolysis around the ring is a feasible explanation of the data. Given all the available experimental evidence, it is our opinion that AAA+ ATPase systems should be versatile enough to allow for the coexistence of multiple pathways of coordinated ATP hydrolysis, and that the existence of a rigorously sequential *Mode 3* hydrolysis does not necessarily exclude the occurrences of ATP hydrolysis that are less sequential and more randomized or of mixed modes. The mode of the ATP hydrolysis is likely energetically dependent on the interaction with specific substrates, regulatory subunits, chaperones or cofactors, which deserve further investigation.

## 4. Mechanistic Variation in Other AAA+ ATPases

### 4.1. Cdc48/p97

How the Cdc48 complex processes ubiquitin-conjugated substrates with two ATPase rings has long remained elusive. Structural studies of Cdc48 in the absence of a substrate showed that nucleotide binding leads to conformational rearrangements of the D1 and D2 rings [101,148]. A cryo-EM structure of an archaeal Cdc48 complex unfolding other Cdc48 copies presented a polypeptide chain inside its translocation channel [131]. Although this condition may not be necessarily physiological, a translocation mechanism resembling *Mode 3* of the proteasomal ATPases was proposed [131]. However, in the eukaryotic Cdc48 complex, only D2 contains the canonical aromatic pore loop residues [149,150]. Hence, the evolutional elimination of aromatic residues in D1 pore loops implies a possibly different translocation mechanism from that of archaea [151].

The recent cryo-EM structure of the yeast Cdc48 in a complex with its heterodimeric cofactor Ufd1-Npl4 (UN) and a polyubiquitinated substrate provides important insights into its mechanism of substrate processing [152] (Figure 5a). The substrate is first recruited by binding of the conjugated polyubiquitin chains to the UN, which binds to the N domains of Cdc48 and acts as the ubiquitin receptor for the Cdc48–UN complex. The ubiquitin chain with at least five ubiquitin moieties is required to achieve optimal binding [153,154]. Surprisingly, a ubiquitin molecule is partially unfolded and traverses a conserved groove of Npl4 all the way through both ATPase rings. How the unfoldase formed by Cdc48 and UN initiates its substrate processing is still unknown. But the unfolding is speculated to start on a segment of the proximal ubiquitin molecule, and initial unraveling must be facilitated by the D1 domains and/or the N domains after ATP-independent insertion of the ubiquitin N-terminus into the central pore [76,152,154]. Biochemical assays have suggested that the functional behavior of the D1 domains in substrate unfolding are different from those of D2. Except for the positional control of the N domains, the D1 domains seem to contribute much less to substrate processing than D2, because ATP hydrolysis by the D1 domains was rarely observed during the substrate processing, whereas the D2 domain actively hydrolyzes ATP in the same time interval [154,155,156,157]. Nevertheless, ATP hydrolysis in D1 does affect the ATPase activities of D2, and also plays an important role in substrate binding and release from the complex [154,158,159,160]. These observations are compatible with the structure of the substrate-bound Cdc48 complex [146,152]. The D1 and D2 domains are connected by the D1–D2 linker region, serving as a flexible hinge. In the D1 ring, despite ATP binding to all ATPase subunits, the unfolded ubiquitin polypeptide interacts with only two ATPase subunits, with the entire D1 ring adopting an almost planar arrangement. In contrast, nucleotide binding and substrate engagement with pore loops in the D2 ring exhibited an architecture highly similar to *Mode 2* of the 26S proteasome, where two adjacent subunits are disengaged from the substrate gripped by a helical staircase of four D2 pore loops (Figure 5b,c). Although no alternative high-resolution structures were obtained to define actual conformational changes of the tandem ATPases during its catalytic cycle, the observation that substrate processing starts from a substrate-linked ubiquitin explains why the Cdc48-UN complex can unfold well-folded compact globular structures without an unstructured N- or C-terminal segment. Interestingly, the Cdc48 complex shows quite different architectures under varying biochemical conditions. For example, in the conformations in the presence of ADP/BeF_x_, the D1 pore loops also arrange as a staircase and the D2 ATPases turn into *Mode 3* with only one D2 domain disengaging from the substrate [146,152]. Despite these biochemical and structural analyses, the mechanisms of substrate selection and termination of the translocation by Cdc48, as well as its connection with the proteasome remain elusive and await further investigation.

### 4.2. FtsH-Like Mitochondrial Proteases

To understand the organization of mitochondrial proteases, several crystal structures of FtsH, the evolutionarily related protease in bacteria, have been reported, which included only the soluble cytosolic region comprising the AAA+ and protease domains. These structures revealed distinct rotational symmetry depending on nucleotide conditions. The nucleotide-free conformation showed a six-fold symmetry [161], while the fully ADP-loaded structures exhibited two-, three-, or six-fold symmetries [162,163,164,165]. To examine the native states, a cryo-EM structure of full-length yeast m-AAA protease with ATP bound at subnanometer resolution was acquired, which is fully compatible with crystal structures, confirming its broad structural resemblance to the bacterial FtsH [166].

To understand the translocation mechanisms of the FtsH-like proteases, cryo-EM structures of the substrate-bound yeast i-AAA Yme1 and human m-AAA AFG3L2 were solved at near-atomic resolution [132,133]. In both studies, the N-terminal transmembrane domains of all six subunits were genetically substituted with a soluble hexameric coiled coil to ensure both hexamerization and degradation activity in vitro. The engineered Yme1 and AFG3L2 complexes revealed conformations of substrate-bound ATPase ring closely resembling *Modes 3* and *2* of the 26S proteasome, respectively (Figure 6b). Nucleotide states allosterically regulate conformations of the entire subunit, which is consistent with the crystal structures with open and close conformations [165]. However the two proteases bear some differences in structural features. Inside the axial channel, the pore-1 loops of AFG3L2 form a spiral staircase with a tightly packed residue that surrounds the substrate, and the pore-2 loops and the central protease loops are closer to the substrate than in Yme1 (Figure 6b). Another apparent difference is the C-terminus. In Yme1, it is an unstructured loop extending away from the protease ring into the mitochondrial intermembrane space. In contrast, the C-terminus of AFG3L2 extends upward from the base of the protease domain along the exterior surface of the complex to the membrane-proximal face with a highly charged tail. The differences in C-termini may help to explain their specificity in substrate recognition and processing [167,168]. However, the unknown structures of the transmembrane domains and their functional role need further investigation.

## 5. Perspectives

In this review, we have discussed typical AAA+ ATPases in three families belonging to the classic clade, which are involved in diverse protein degradation pathways. These AAA+ ATPases assemble into the ring-like architecture of hexameric complexes and share the presumably similar nucleotide-driven mechanisms in substrate unfolding and translocation. The most frequently proposed mechanism hypothesizes a sequential cycling of ATP hydrolysis unidirectionally around the ATPase ring [11,118,132,145]. However, due to limitations of the current techniques in molecular and structural biology, it has been very difficult to determine all intermediate steps in detail. Indeed, in most structural studies on the substrate-bound AAA+ ATPases, only one high-resolution conformation was determined under a given biochemical condition, and the associated proposal of the translocation mechanism was largely hypothetical [131,132,133,138,146]. This means that the sequential hydrolysis model may not necessarily account for all translocation activities in vivo, given that intracellular biochemical compositions and signals are highly heterogeneous and noisy. To date, the greatest number of distinct AAA+ ATPase conformations analyzed (more than 20) is based on the human 26S proteasome, which provides the most extensive picture of AAA+ ATPase dynamics [11,51,52,53,59,116]. The differential ubiquitin–substrate densities intrinsic to some of these structures are the only available pieces of evidence for the time sequence of the corresponding conformations along the pathway of chemical reactions [11]. Remarkably, three coexisting modes of coordinated ATP hydrolysis were associated with ubiquitin recognition, deubiquitylation, translocation initiation and processive degradation in the substrate-bound human 26S proteasome [11]. Each mode of coordinated ATP hydrolysis was also observed in structural snapshots of various AAA+ ATPases under specific biochemical conditions by studies on the ClpX (*Mode 1*) [144], ATG3L2 (*Mode 2*) [133], Cdc48/p97 (*Modes 2* and *3*) [146,152], Yme1, ClpB, ClpXP, yeast 26S proteasome and bacterial T7 replisome (*Mode 3*) [118,123,132,147,169,170], suggesting highly conserved dynamic patterns in the structure–function relationships of AAA+ ATPase hexamers. These mechanistic findings, especially the key features and interactions in the high-resolution structures, are expected to facilitate pathological studies of the AAA+ proteases, as well as therapeutic development and drug discovery for regulating proteolysis effects in treating various diseases. Future structural studies are expected to be directed toward resolving key intermediate states necessary for unambiguously defining the detailed mechanisms of substrate processing by each of those AAA+ ATPases, as well as how these intermediate states are associated with their diverse functioning in vivo.

## Figures and Tables

**Figure 1 biomolecules-10-00629-f001:**
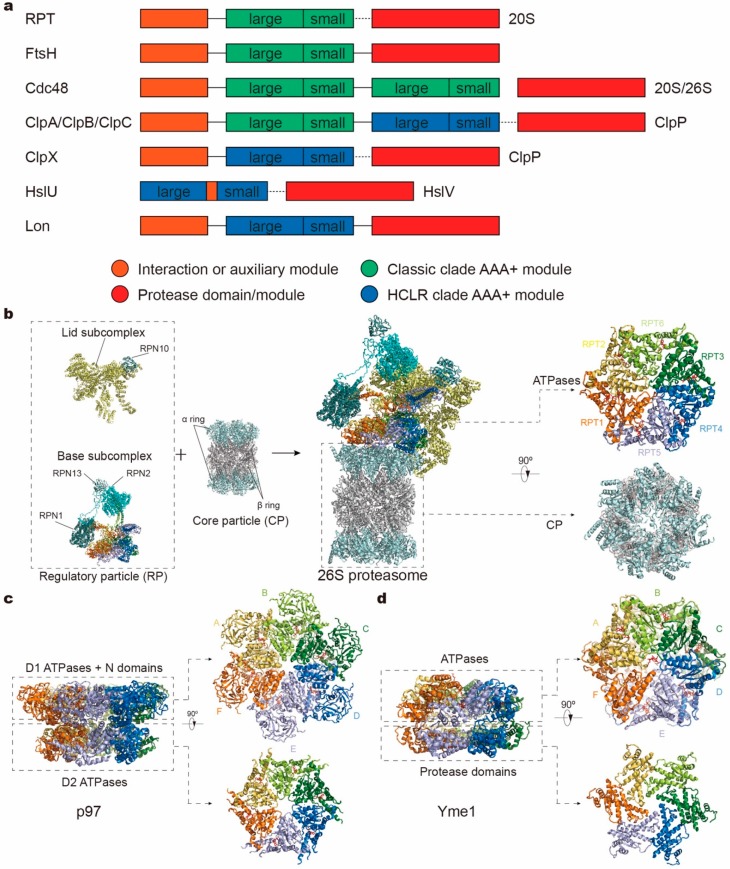
Domain organizations and structures of protease complex machineries of adenosine triphosphatases (ATPases) associated with a variety of cellular activities (AAA+). (**a**) Domain organizations of AAA+ proteases in different families. The length of the bar is not linearly proportional to the real length of the corresponding sequence. Each protein contains one or two AAA+ modules, each consisting of a large and small subdomain, and additional family-specific domains, which are not specifically depicted here. Protease modules reside in separate protein subunits except for FtsH and Lon. Protein subunits connected by dotted line can assemble into one complex. (**b**–**d**) Atomic models of the yeast 26S proteasome (**a**; PDB ID: 6FVT), the ADP-bound human p97 (**b**; PDB ID: 5FTK) and the substrate-bound yeast Yme1 (**c**; PDB ID: 6AZ0) as the representative AAA+ proteases. Orthogonal views of their proteolytic complexes/domains and hexameric ATPase rings are shown here.

**Figure 2 biomolecules-10-00629-f002:**
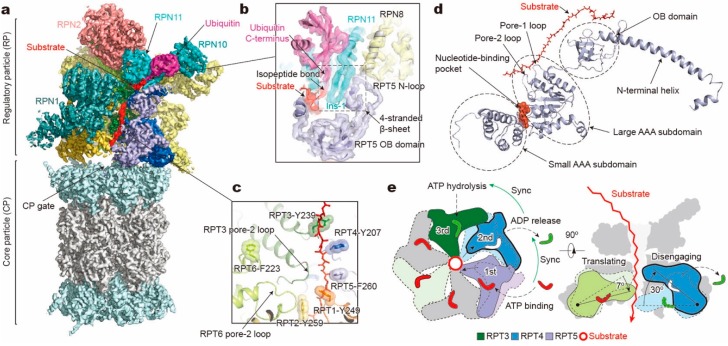
Structure and translocation mechanism of the human 26S proteasome. (**a**) Cryo-EM structure of the substrate-bound human proteasome in state E_B_ at 3.3 Å (EMDB ID: 9218; PDB ID: 6MSE). The RPT1 density is omitted to show the substrate density inside the ATPase ring. The RPN13 density is not observed in this structure. (**b**) A close-up view of the quaternary interface around the isopeptide bond between substrate and ubiquitin. (**c**) Architecture of pore loop staircase interacting with the substrate. Aromatic residues in pore-1 loops are labelled. (**d**) Molecular model of RPT5 in state E_B_, with ATP bound and substrate engaged. (**e**) Schematic of mechanical substrate translocation of proteasomal ATPases. Synchronization of nucleotide processing in three adjacent ATPases (left) causes differential vertical rigid-body rotations in each substrate-engaged ATPase that cooperatively transfer the substrate (right).

**Figure 3 biomolecules-10-00629-f003:**
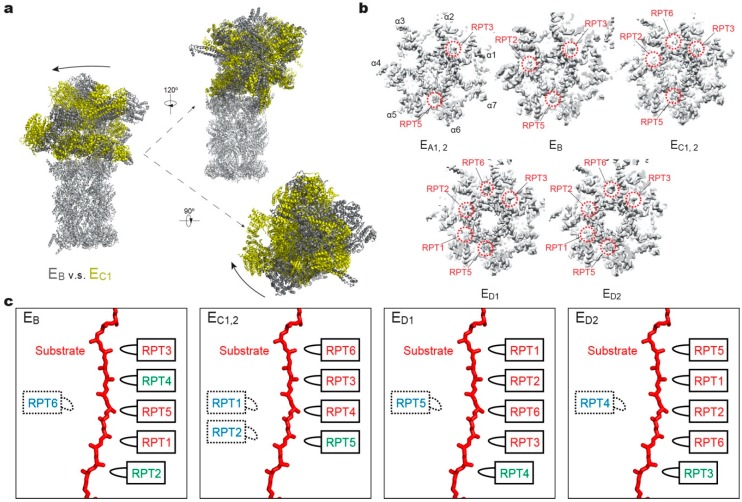
Structural features of the substrate-bound human 26S proteasome in different conformational states. (**a**) Conformational switching of the RP during the transition from state E_B_ (dark gray; PDB ID: 6MSE) to E_C1_ (yellow; PDB ID: 6MSG), with the CP (gray) aligned. (**b**) Cutaway surface representations of the RP–CP interface in different states. The red dashed circles highlight the densities of the RPT C-terminal tails that are inserted into the α-pockets of the CP. (**c**) Schematics showing the relative locations of the pore-1 loops of six RPT subunits along the vertical axis. RPT subunits in ATP-bound (red), ADP-bound (green) and apo-like (blue) states are depicted alongside the substrate from top to bottom and subunits that are disengaged from the substrate are placed on the left side. States E_A1_, E_A2_ and E_C1_ are omitted here, as their ATPase structures are identical to that of E_B_ and E_C2_, respectively.

**Figure 4 biomolecules-10-00629-f004:**
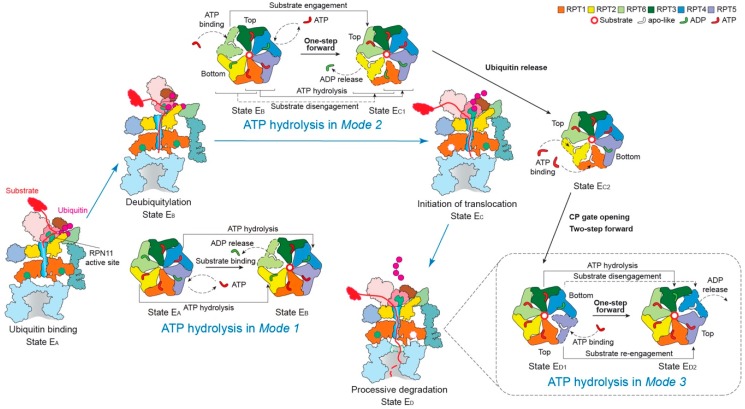
Schematic of coordinated ATP hydrolysis and nucleotide exchange observed in seven states in the substrate-bound human 26S proteasome [11]. Three principal modes are depicted here, with *Modes 1*, *2*, and *3* featuring hydrolytic events in two oppositely positioned ATPases (yellow and blue), in two adjacent ATPases (orange and violet) and in one ATPase at a time (forest green), respectively. The RPT subunits with their pore-1 loops on the top and bottom of the pore-loop staircase are labeled “Top” and “Bottom”, respectively, which are consistent with Figure 3c.

**Figure 5 biomolecules-10-00629-f005:**
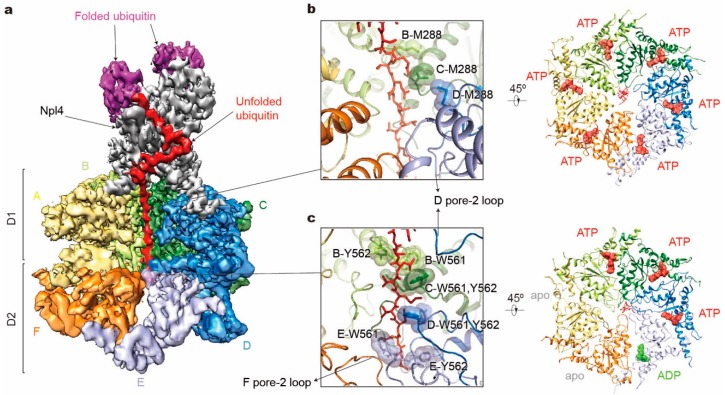
Structure illustration of the tandem hexameric ATPases of Cdc48 processing a substrate. (**a**) Cryo-EM density map of the Cdc48– Ufd1-Npl4 (UN)–substrate complex in the presence of ATP and with Cdc48 carrying a Walker B mutation (EMDB ID: 0665; PDB ID: 6OA9). D1 domains of subunits E and F are both omitted for clarity. Densities of Ufd1 and N domains are too vague to be shown. (**b,c**) Substrate interactions with the Cdc48 D1 pore (**b** left) and D2 pore (**c** left), and corresponding nucleotide states of six ATPases (right). Only key residues in pore-1 loops are labeled here.

**Figure 6 biomolecules-10-00629-f006:**
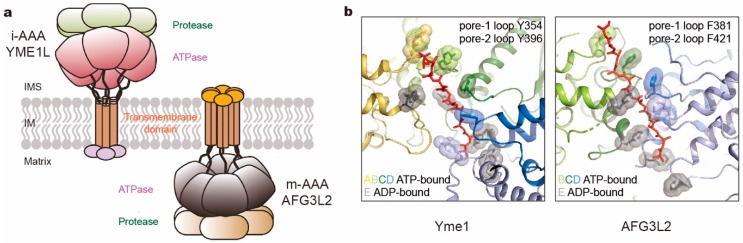
Comparison of two kinds of mitochondrial membrane-anchored AAA+ proteases. (**a**) Cartoon representation of i- and m-AAA proteases anchored in the mitochondrial inner membrane (IM) with opposite orientations to intermembrane space (IMS) and matrix space, respectively. (**b**) The pore loop spiral staircases surrounding the substrate in yeast Yme1 (PDB ID: 6AZ0; left) and human AFG3L2 (PDB ID: 6NYY; right). Aromatic residues in pore loops are all highlighted as sticks and spheres, which in pore-2 loops are colored gray.

**Table 1 biomolecules-10-00629-t001:** Subunits and function of the human 26S proteasome.

Subunits	Function
RP lid subcomplex	
RPN3, RPN5, RPN6, RPN7, RPN9, RPN12, DSS1	Structural
RPN8	Dimerization with RPN11 MPN
RPN11	Deubiquitinase
RPN10	Ubiquitin receptor
RP base subcomplex	
RPN1	Ubiquitin receptor/USP14 binding
RPN2	Structural
RPN13	Ubiquitin receptor/UCH37 binding
RPT1-RPT6	AAA+ ATPase
CP subcomplex	
α_1_-α_7_	CP gate
β_1_, β_2_, β_5_	Proteolytic chamber, peptide hydrolysis
β_3_, β_4_, β_6_, β_7_	Proteolytic chamber
Non-stoichiometric	
USP14, UCH37	Deubiquitinase

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
