# Peer review of "AAA+ ATPases in Protein Degradation: Structures, Functions and Mechanisms"

_biomolecules, 2020, doi:10.3390/biom10040629_

Round 1

Reviewer 1 Report

The manuscript presents an in depth, high resolution analysis  of the structure, conformational changes and mechanism of action of three AAA+ATPase families as emerged from recent cryogenic electron microscopy studies.

The review is interesting but also highly specialized. In the present form the content seems to be more suitable to the experts in the sector than to the general audience. Some effort is necessary to expand the readability and usability of the review. In particular:

1) More emphasis should be given to the physiological relevance of the novel acquisitions, and their implication in pathological conditions.   

2) A schematic picture should be added to better illustrate the general concept of the proteases mode of action expressed in lines 36-48, which is the basis to understand what is described in the next sections.

3) Certain technical terms are first mentioned without explaining their meaning see in example  “clade” (line 50), “presensor-1 insertion” (line 54),  or “state” (line 229).

4) In the caption to fig. 3: the authors state “Three principal modes are depicted here, with Modes 1, 2, and 3 featuring hydrolytic events in two oppositely positioned ATPases, in two adjacent ATPases and in one ATPase at a time, respectively”. I suggest to add in the caption also the colors that distinguish the different ATPase modes of action, i.e. : “Three principal modes are depicted here, with Modes 1, 2, and 3 featuring hydrolytic events in two oppositely positioned ATPases (yellow and blue), in two adjacent ATPases (orange and violet) and in one ATPase at a time (green and violet), respectively”.

Author Response

Dear Reviewer,

Thank you very much for constructive comments. We have addressed all the suggestions, as shown in the point-by-point response below.

Comments: 1) More emphasis should be given to the physiological relevance of the novel acquisitions, and their implication in pathological conditions

Response: We have revised the first paragraph and the last section to include a brief description to underscore the physiological relevance and their pathological implications. See lines 31-35 and 531-534.

Comments: 2) A schematic picture should be added to better illustrate the general concept of the proteases mode of action expressed in lines 36-48, which is the basis to understand what is described in the next sections.

Response: We have revised Figure 1 to include three additional panels illustrating the general concept of the protease architecture. See page 3.

Comments: 3) Certain technical terms are first mentioned without explaining their meaning see in example “clade” (line 50), “presensor-1 insertion” (line 54), or “state” (line 229).

Response: We have added explanations for these terms. See line 53-54, 56, and 248.

Comments: 4) In the caption to fig. 3: the authors state “Three principal modes are depicted here, with Modes 1, 2, and 3 featuring hydrolytic events in two oppositely positioned ATPases, in two adjacent ATPases and in one ATPase at a time, respectively”. I suggest to add in the caption also the colors that distinguish the different ATPase modes of action, i.e. : “Three principal modes are depicted here, with Modes 1, 2, and 3 featuring hydrolytic events in two oppositely positioned ATPases (yellow and blue), in two adjacent ATPases (orange and violet) and in one ATPase at a time (green and violet), respectively”.

Response: This was done. See Line 615-654.

Reviewer 2 Report

In this review, the authors review the functions of AAA+ ATPases involved in protein degradation. The review is comprehensive, well-written and organized. I think it will be of interest to a broad audience.

Minor points to improve the manuscript

  1. In the introduction, the authors indicate that there are the clades (line 52). While I understand the details may be out of the scope of this review, I think the differences/basis of this division should be described, at least succinctly.
  2. Section 2: The 26S proteasome section is difficult to get through; I recognize that there are many subunits and this contributes to complexity. I wonder if this section might be improved by adding a table of different subunits, their functions and positions within the overall proteasomal structure?
  3. Overall organization: I felt like the review would benefit from combining sections 2 and 3 and dealing with one example at a time (ie. Combine 2.2 and 4.1, 2.3 and 4.2) and so on. It gets a little confusing to go back and forth.
  4. Sections 3.1 to 3.5- I feel like more schematics would help understand the workings of the 26S proteasome much better. While figure 3 is useful, I wonder if the authors can add more schematics for this section, particularly when describing the organization of the different subunits within the proteasome.
  5. Figure 3: The labeling here is a bit confusing, what do the authors mean by “top” or “bottom”?

Author Response

Dear Reviewer,

Thank you very much for constructive comments. We have addressed all the suggestions, as shown in the point-by-point response below.

Comments: In the introduction, the authors indicate that there are the clades (line 52). While I understand the details may be out of the scope of this review, I think the differences/basis of this division should be described, at least succinctly.

Response: We have added explanations of how the clades are categorized. See line 53-54.

Comments: Section 2: The 26S proteasome section is difficult to get through; I recognize that there are many subunits and this contributes to complexity. I wonder if this section might be improved by adding a table of different subunits, their functions and positions within the overall proteasomal structure?

Response: We have added Table 1 to address this suggestion. See page 4.

Comments: Overall organization: I felt like the review would benefit from combining sections 2 and 3 and dealing with one example at a time (ie. Combine 2.2 and 4.1, 2.3 and 4.2) and so on. It gets a little confusing to go back and forth.

Response: We have significantly expanded Figure 1 and added a number of panels to illustrate the similarity of three systems. Thus, section 2 is dedicated to briefly introduce each system and show the common aspects of their architectures, whereas section 4 is focused on discussing the differences and variations. The revised Figure 1 should help readers to avoid going back and forth when they go through the subjects.

Comments: Sections 3.1 to 3.5- I feel like more schematics would help understand the workings of the 26S proteasome much better. While figure 3 is useful, I wonder if the authors can add more schematics for this section, particularly when describing the organization of the different subunits within the proteasome.

Response: We have added a new Figure 3 on page 8 to address this suggestion.

Comments: Figure 3: The labeling here is a bit confusing, what do the authors mean by “top” or “bottom”?

Response: We have added a sentence in Figure 4 legend to explain the meaning of “top” and “bottom”. See Line 363-364.

Reviewer 3 Report

Dear Shuwen Zhang, dear Youdong Mao, I read your review about the structure-function relationship of the AAA+ ATPases with great interest.

I think it is very comprehensive and well balanced. While I have no scientific comments, I would have some editing comments that aim to improve readability.

L9 delete "the heart of"

L10 delete "fundamentally" and replace "myriad" with many

L12 replace "through" with inside

L14 replace "accomplished" with resolved L16 replace "recent" with resulting

L17 with an emphasis....analysis of the 26S...

L18 delete "together"

L19 relationship not relationships

L19 ....ATPase hexamers that were also observed in the human 26S proteasome thus suggesting common dynamic models of the mechanochemical coupling during force generation and substrate translocation.

L27 a fundamental role in...

L28 cell cycle regulation

L29-32 Proteolysis affects not only misfolded or otherwise damaged proteins but also regulatory proteins to maintain the function of cellular integrity thereby preventing human diseases.

L32-33 delete "In cells" start with Proteins that are earmarked for degradation are....and are therefore tagged...

L34 from normal cellular constituents (delete those), delete "target" (of these proteins)

L37 (ATPase) of the AAA+ L40 that couples ATP hydrolysis

L42 delete "proteloytic"....such as the proteasome

L43 subunits functions as substrate-remodeling engines

L44 axial pore that guards the entry

L52 delete "mainly" ... is called the classic....including the proteasome family

L54 delete "clade after HCLR

L55 including the HsIU/ClpX

L57 AAA+ module that consits...

L63 these recent results...

L64-65 the labelling in Fifure 1 is a bit confusing as it is unclear how the prorein names on the left and right side refer to the depicted protein structures. I assume the dotted line indicates that, for example the red domain is only present in ClpP but not in ClpA/ClpB/ClpC. Is this correct? I think this should be explained in the figure legend.

L72 in the 26S Proteasome

L73 The ubiquitin-peoteasome....pathway provide the majority of...degradation activity in eukarytic cells

L81 introduce "CP" and "RP" to the reader here as the abbreviations are used for the first time

L82 as the 20S proteasome...

L107 The pore loops interact with...

L109...and sensor 2 domains from one...

L110 two R-finger domains from the large AAA...

L112 should there be a reference to Fig. 2e after ...adjacent subunits [34].?

L124 start the sentence with "The central entry port....

L130 By contrast to what (the relation to what the statement is contrasted with is unclear)

L130 ...seems to contribute...

L1312 In both, the human...

L136 it is unclear what is segregated with regard to the ubiquitinated substrates

L139 assemble with the 20S..

L140 through in vitro crosslinking

L153 to the single AAA domain

L156 the sentence "The N domains are not fixed with respect to the double ring." needs an explanation and/or context

L158 explain why the "upward conformation" is important

L159 like the heterodimer...binds to the N domains

L160 extreme C-termini (of what?)

L161 These cofactors supply Cdc48 with...

L166 within the mitochondria...counterpart in the cytosol

L173 whereas the FtsH-like proteases are uniquely...

L177 specialization of FtsH-like proteases... replace "Despite" with Independently of the...

L189 The structure of the 20S...by X-ray..

L190 In contrast to the high stability

L194 suggest that in vivo the 26S....holoenzymes mostly stay....

L195 absence of a substrate....sample distinct alternative conformations

L198 subunits....inhibitors [117, 118]or by replacing...with the slowly...

L203 suties offer the first

L207 of the human 26S proteasome in a time dependent manner...

L208 heterogeneity of the 26S proteasome

L210 delete "in this case"

L211 extensively using the latest machine-learning tools

L212 eventually obtained seven

L215 delete certain"

L217 conformations of the

L220 across the eukaryotic kingdom

L229 end the sentence after "substrate" and start a new sentence with "One of the key..."

L230 proteasome at the deubiquitinylation step appears to be missing...

L235 replace "mission" with step

L241 "translation abve the CP" is unclear for the reader

L241-242 transition of the CP gate from its closed to its open state. The observation of the Eb state provides information about critical intermediates...

L243 as it allows the observation of the gradual activation of the CP by the stepwise insertion...

L244 delete "one at a time" process of the CP gate opening L245 channel of the RPT ring

L249-250 coupling the ATP hydrolytic cycle with substrate processing

L254 corresponding to a one step translocation driven by...

L255 of a single L256 to be a key structural featrue that is....including the 26S..

L257 ...[129, 130] and Hsp104...

L260 ... underneath the pore-1 loops

L266-268 An ATP molecule within this pocket contacts both AAA domains thus determining the hinge configuration and locking the AAA domain...

L269 outward flip of

L271 of the nucleotide... determines

L272 stats of the nucleotide-binding pocket

L273 and an apo-like state...density is present inside

L277 trigger an intrinsic

L278 into an intra-domain energy potential stored

L280-L281 liberates the stored energy and converts it

L285 model rather than a random

L286 studies of the proteasome and the bacterial

L287 features of this model....subunits make contact

L292 proteasome possesses more inter-subcomplex

L293 delete "On one side" start the sentence with "The ATPase ring..."

L294 in the base, and it forms a multivalent... (delete "On the other side, the ATPase ring")

L299 steps of the proteasome

L308 RPt4 are hydrolyzed

L310 delete "of the AAA ring"

L313 because of the lack

L318 delete "is about to" (the proteasome initiates substrate translocation...)

L319 replace "To this end" with "The latter s achieved when ATP molecules...

L319 the sub-sentence "with RPT1 and its clockwise neighbor" is unclear to the reader

L321 pore loop

L323 ...distance of two residues (of the substrate?) - unclear to the reader

L324 staircase, while RPT5 is about to...

L337 and flips

L338 delete 2anticlockwise adjacent"

L340 was in an apo-like

L343 that are engaged.. delete "approximately"

L344 delete "in concert"

L345 translocation has been observed

L348 replace "observed" with detected

L350 delete "labeling the time sequence"

L351 delete "in this study"

L359 delete "in these structural studies" delete "several"

L360 delete "unambiguous" ...features that revealed the time sequence of the events along the pathway

L363 what means "common buffer" in this context

L364 delete "along the pathway" ..of the chemical reactions

L365 replace "navigating" with moving through all six ATPases...

L372 replace "By comparison" with Interestingly, ...but lacks the RPN11-bound ubiquitin and the nucleotide density in RPT1 thus verifying...

L374 delete "Similarly" start the sentence with Both states, Ed1 and Ed2, show...

L376 unclear what "toggling of RPN11 Insertion-1 motif" means ... these structures suggest a spatiotemporal continuum providing us with the only direct...

L379 delete "However" start the sentence with One should note however that...

L381-382 This leaves the possibility open that a sequential Mode-3 hydrolysis around the ring is a feasible explanation of the data.

L382 delete "speculation and"

L385-387 delete "possible" .. are less sequential and more randomized or of a mixed mode. The mode of ATP hydrolysis is likely... dependent on the interaction with...

L388 delete "in-depth"

L392 absence of a substrate showed that nucleotide

L396 delete "proteolysis in vivo"

L397 replace "for eukaryotic Cdc48" with in the eukaryotic Cdc48 protease only D2....

L398 Hence, the evolutional.... in the D1 pore loop (delete eukaryotic)

L407 The recent ...

L408 replace "essential" with important ... its mechanism of substrare...

L413 please introduve the protein UN to the reader

L416 that the functional behavior of the..

L417 delete "obviously"

L419-420 was rarely observed during substrate processing... hydrolyzes ATP in the same time interval

L423 structure of the substrate-bound

L424 connected by the D1-D2..

L431 replace "functioning" with catalytic cycle ... that substrate processing

L434-435 For example, the conformations....and the D2..

L438 selection and termination of the translocation by Cdc48, as well as its connection with the proteasome remain elusive...

L441 delete "assembly and"

L444-445 conditions while the nucleotide-fee.....[158], the fully ADP-loaded...

L446 symmetries ... To eamine the native states L461 ring closely resembling ... delete "respectively"

L464 ...bear however some structural differences.

L465 surrounds L466 to the substrate

L468 introduce "IMS" to the reader here

L470 of the C-termini may help to explain

L478 replace "hypothesize" with propose a sequential...

L480 all intermediate steps...

L482 delete "experimentally" ... under a given biochemical condition

L483 of the translocation mechanism.... This leaves the gate open for a model such that the sequential.....

L485 delete "up to"

L486 than 20) is based on the human 26S proteasome, thus providing the most...

L488 structures are the only... delete "labeling"

L490 of a coordinated delete "observed to be"

L493 replace "similarly" with also

L497-498 suggesting either a divers structural pattern of different modes in different AAA+ ATPase hexamers or o combined mode of which only individual sub-stats were captured in the given structures.

L498 delete "additional"

L499 delete "during the complete cycle

L500 their divers functions in vivo

Author Response

Dear Reviewer,

Thank you very much for constructive comments. We have revised the manuscript by taking all suggestions you described. We have provided a tracked version of the manuscript where all changes are marked and tracked.

Round 2

Reviewer 1 Report

The authors have satisfactorily responded to all my comments.